

# Design of judicial public opinion supervision and intelligent decision-making model based on Bi-LSTM

Heng Guo

College of Humanities, Law and Foreign Languages, Taiyuan University of Technology, Taiyuan, Shanxi, China

## ABSTRACT

Fuzzy preference modeling in intelligent decision support systems aims to improve the efficiency and accuracy of decision-making processes by incorporating fuzzy logic and preference modeling techniques. While network public opinion (NPO) has the potential to drive judicial reform and progress, it also poses challenges to the independence of the judiciary due to the negative impact of malicious public opinion. To tackle this issue within the context of intelligent decision support systems, this study provides an insightful overview of current NPO monitoring technologies. Recognizing the complexities associated with handling large-scale NPO data and mitigating significant interference, a novel judicial domain NPO monitoring model is proposed, which centers around semantic feature analysis. This model takes into account time series characteristics, binary semantic fitting, and public sentiment intensity. Notably, it leverages a bidirectional long short-term memory (Bi-LSTM) network (S-Bi-LSTM) to construct a judicial domain semantic similarity calculation model. The semantic similarity values between sentences are obtained through the utilization of a fully connected layer. Empirical evaluations demonstrate the remarkable performance of the proposed model, achieving an accuracy rate of 85.9% and an F1 value of 87.1 on the test set, surpassing existing sentence semantic similarity models. Ultimately, the proposed model significantly enhances the monitoring capabilities of judicial authorities over NPO, thereby alleviating the burden on public relations faced by judicial institutions and fostering a more equitable execution of judicial power.

## INTRODUCTION

China's judicial organs mainly refer to the people's courts and people's procuratorates. In a broad sense, public security is also considered as a part of the judicial organs. Taking the "Procuratorial Affairs Publicity" carried out by the people's Procuratorate as an example, it refers to the activities and matters related to procuratorial functions and powers disclosed by procuratorial organ to the public according to law. The state and the Supreme People's Procuratorate successively issued a series of documents and regulations, establishing and improving the system construction and process design of procuratorial affairs publicity. Since 2014, through the "People's procuratorate case information disclosure network", the national "procedural information inquiry", "important case information" and "legal

Corresponding author
Heng Guo, guoheng268@163.com

documents" have been made public in accordance with the law, and hundreds of millions of Chinese Internet users have the opportunity to access and supervise the work of the people's procuratorate through portals, microblogs, wechat, news clients and other channels (*Hu, 2019*). Similar work of information disclosure is also carried out in the people's government and the people's court according to law. But what can not be ignored is that some cases and related documents online public opinion, especially the negative impact, has become a practical problem that the judicial organs must face. Vicious public opinion pressure on the judiciary will interfere with the principle of independent trial, which even causes the judiciary to be under pressure to selectively disclose information, which is contrary to the original intention of information disclosure (*Shammi et al., 2021*). We must realize that in the era of open information, independent trial and respect for public opinion are not contradictory. If there is the support of public opinion monitoring technology, it can greatly reduce the pressure of public opinion of procuratorial and judicial organs, and promote the fair and fair exercise of power.

In today's global culture, the ability to analyze and predict citizen behavior at scale has become a priority for governments when it comes to safety regulation. *Saura, Palacios-Marqués & Iturricha-Fernández (2021)* used data mining technology to classify and discuss the risks to citizens' privacy based on the types of AI strategies used by governments that may affect collective behavior and lead to large-scale behavior modification. A network public opinion (NPO) event refers to any incident, activity, or topic that arises and spreads through online networks, capturing the attention and engagement of Internet users. These events can range from trending news topics and social issues to crises and controversies, all of which become focal points of public discourse within digital environments. The NPO monitoring system was the public opinion analysis software in the early stage, and its function was very limited. Nowadays, various intelligent identification technologies and data analysis and mining technologies are developing rapidly. The public opinion monitoring system has already bid farewell to the stage of manual detection, screening, analysis and prediction, and has developed into a NPO monitoring system with complete functions and advanced algorithms, which mainly involves network information collection technology, topic detection and tracking technology, text sentiment analysis technology and other technologies (*Wang, 2021*). *Saura, Ribeiro-Soriano & Palacios-Marqués (2022)* use the LDA topic model to analyze the relevance of user privacy data based on data-driven innovation, which promotes the development of complex government strategies for data management, user behavior prediction or user behavior analysis. However, the public opinion monitoring oriented to the judicial field is more subdivided, and its particularity mainly lies in the scope of event monitoring and some data processing methods. During the period of the project, we found that the automation and intelligence of such systems for judicial organs in China are relatively deficient, for example, in monitoring, it is necessary to define keywords manually, which can not automatically identify events, resulting in low efficiency; for massive data, only statistical analysis visualization is carried out, and semantic analysis technology (or only a small amount of technology such as emotional analysis) is not fully utilized (*Chen et al., 2021*; *Ma & Liu, 2014*).

The main contribution of this study is to the legal domain knowledge combined with social media, from the perspective of legal elements to combine domain knowledge into text vector, build the judicial field of public opinion monitoring model based on semantic features analysis, the model could find similar cases from the legal elements of the level, effectively improve the judicial regulation ability of network public opinion, greatly alleviate the public relations pressure of judicial organs, so that judicial organs more fair and equitable exercise of power.

The model constructed in this study comprehensively considers the characteristics of time series, binary semantic fitting and public emotion strength, and constructs a judicial semantic similarity calculation model based on Bi-LSTM network (S-Bi-LSTM). Finally, it runs circularly through a single scheduler process to achieve effective judicial discrimination.

# RELATED WORK

## NPO monitoring technology

The most typical carrier of online public opinion is social network. Typical social networks include foreign Facebook, twitter and other platforms, as well as domestic microblog platforms. The network is generally presented in the form of graph, while the user is abstracted as a node. The information is transmitted through the edge between nodes. The dissemination of public opinion information on social networks can be quantified by some structured network data, such as the number of likes, the number of forwarding, the number of comments and so on. Through the use of appropriate technical analysis tools and visualization technology, we can depict the dissemination situation of specific information and follow-up tracking.

There are many kinds of network information, and the audience's ability to receive and digest information is limited. Therefore, most of the network information will not attract the public's sustained and deep attention, that is, it will not develop into NPO. The development of technology has changed the content of information. On the one hand, the form of information dissemination is not only limited to simple words, but also combined with pictures, hyperlinks, voice and short videos. Different presentation forms increase the attraction and communication power of the content. On the other hand, NPO is affected by the sensitivity of content and the explosive degree of events (*Li & Wang, 2021*). If public opinion events are related to the social needs or interests of the public, and are more likely to resonate with the needs of the public, then the content of public opinion itself has more communication ability and is more likely to trigger public discussion and communication (*Shi & Wang, 2016*).

Nowadays, in the dissemination of public opinion on social network, not only text information can be extracted, but also the emotion of text can be extracted from the network. *Liu (2010)* analyzed the NPO under the micro content. They defined the micro content as the combination of information ontology and information state, and analyzed the information characteristics, opinion convergence and evolution characteristics of micro content. In addition, the formation of hot spots is the main reason for the formation of NPO due to the fusion convenience, explosibility and sociality of micro content

(*Liu, 2010*). *Gutierrez & Sequeda (2021)* conducted a lot of scale analysis and social network analysis by using Co-word analysis and visualization methods, using the knowledge of knowledge map in the hot spots of frontier research literature. *Mao et al. (2021)* introduced the related technologies of ontology and semantic computing into the topic discovery research of network group events, and carried out an empirical study. The trial results demonstrate the way that the strategy can really obtain topic information, which is helpful to the topic discovery of network group events (*Mao et al., 2021*). *Saura, Ribeiro-Soriano & Palacios-Marqués (2021)* From the perspective of social media ethical design and monitoring capitalism, this article uses social networks to increase user engagement and modify user online behavior by developing a systematic literature review.

Trend prediction of NPO can be divided into quantitative prediction and qualitative prediction. Qualitative prediction only needs to predict whether the trend is going up or down in the future, while quantitative prediction needs to predict specific values. In terms of trend prediction, there are currently methods based on machine learning; for example, *Zeng (2014)* used SVM combined with grey prediction to predict public opinion, *Wei, Chen & Zheng (2015)* integrated chaos theory and improved radial basis function neural network to predict public opinion, *You & Chen (2016)* used improved particle swarm optimization combined with BP neural network to predict public opinion, and *Qian, Tao & Lü (2014)* improved the performance of PageRank by adding background nodes based on page rank algorithm.

It can be concluded that the algorithm of public opinion based on large scale and not suitable for network monitoring can also be obtained based on the strong monitoring characteristics of the network. Therefore, we cannot accurately judge the trend of public opinion, nor can judicial department we strictly control and punish the vicious public opinion.

## Semantic similarity

In natural language, there are very complex relationships between words. In specific applications, a simple quantity is needed to measure this complex relationship, and semantic similarity is one of them. The core issue of this study is to quantify the similarity between judicial cases and hot topics of public opinion. *Chatterjee & Gupta (2019)* proposed a sentence similarity measurement method based on linear model considering syntactic and semantic features of sentences, which was well applied to machine translation tasks; *Farouk (2018)* proposed a method combining the pre training word vector and WordNet to measure the semantic similarity between two sentences.

Although the traditional sentence semantic similarity calculation method is effective, it needs to construct feature engineering manually. In recent years, sentence semantic similarity algorithm based on neural network can calculate the semantic similarity between two sentences without manual extraction of text features, and has achieved better results. *He, Gimpel & Lin (2015)* used CNN to extract multi angle features of sentences, and then used these features to calculate the similarity between sentences; *Mueller & Thyagarajan (2016)* used LSTM to represent the sentence vector, and calculated the Manhattan distance between the sentence vectors to obtain the semantic similarity value of two sentences;

*Zhuang & Chang (2017)* combined the BGRU with the attention mechanism to extract semantic features of sentences. Meanwhile, the cosine similarity value of word pairs in parallel sentences was calculated and used as auxiliary features. The sentence vector and auxiliary feature vector were input into the multi-layer perceptron. The similarity score of the two sentences was obtained (*Zhuang & Chang, 2017*).

The rise of pre training models has improved the performance of almost all NLP tasks. *Peters, Neumann & Iyyer (2018)* proposed a context sensitive word representation named Elmo, which is derived from the pre training language model and can be easily used in other tasks. The model achieves optimal performance in six natural language processing tasks including sentiment analysis and text classification for Stanford sentinel tree Bank (SST). In the same year, *Devlin et al. (2018)* proposed a new word representation model, Bert, which overcomes the limitations of the traditional one-way language model, and also achieves optimal performance in 11 natural language processing tasks including sentiment analysis and text classification.

However, these methods ignore the use of legal knowledge, and the dissemination of public opinion based on social media has its particularity. Therefore, this article combines domain knowledge into the vector representation of text from the perspective of legal elements, so that the model can find similar cases from the level of legal elements, in order to make better judicial decisions.

# NPO MONITORING MODEL IN JUDICIAL DOMAIN BASED ON SEMANTIC FEATURE ANALYSIS

The problem of monitoring NPO is that for a given text about a specific network information, the task of the current system is to determine whether the information is public opinion which can be defined as a binary classification problem. By inputting tweets, the tweets are classified into public opinion information and normal information.

## Time series model

This section introduces a time series model specifically designed for analyzing NPO, which is represented as a nonlinear feature sequence. The model aims to capture the dynamic and complex nature of NPO through sophisticated feature extraction and clustering techniques.

Set the d-dimensional random variable $U_i$, and the monitored network public opinion $U = \{U_1, U_2, \cdots, U_N\}$, represents the monitored network public opinion data across different time points. These variables are subjected to feature extraction and cluster analysis on a network platform to identify patterns and trends within the NPO data.

The propagation of NPO across different domains can be captured by analyzing the inter-domain association characteristics at the routing link layer. This is mathematically expressed as:

$$p(U/\Theta) = \sum_{k=1}^{K} \alpha_k G\left(U \mid u_k, \sum k\right) \tag{1}$$

where: $\alpha_k$ represents the baud rate, which reflects the rate at which data related to public

opinion is transmitted or processed. $u_k$ denotes the monitoring frequency response, indicating how often or how sensitively the system monitors changes in public opinion. These variables are critical as they define how quickly and accurately the model can capture the evolving nature of public opinion on the network.

The model also introduces a measure of the comprehensive relative closeness degree of NPO, which can be expressed as:

$$\Delta : [0, T] \to S \times [-0.5, 0.5] \tag{2}$$

Then, the semantic features of public opinion are then further decomposed using a binary semantic decomposition process:

$$\Delta(\beta) = \begin{cases} s_k, & K = \text{round}(\beta) \\ a_k = \beta - K, & a_k \in [-0.5, 0.5] \end{cases} \tag{3}$$

where: $\Delta(\beta)$ defines the relationship between time $\beta$ and the semantic characteristics of NPO. $S$ represents the evaluation set of hotspot times—moments when public opinion spikes. $K$ is the keyword extraction operator, which identifies critical keywords within the NPO data. $s_k$ refers to the k-th element in the evaluation set $S$, representing specific semantic elements identified within the NPO. $a_k$ is the binary semantic feature associated with $s_k$, taking values within the interval $[-0.5, 0.5]$.

The frequently occurring vocabulary on the network platform is screened. When the inverse function satisfy the Formula (4). The semantic features of public opinion are decomposed by Fourier transform and transformed into Formula (5):

$$\Delta^{-1} : S \times t[-0.5, 0.5] \to [0, T]. \tag{4}$$

To further enhance the model, a Fourier transform is applied to decompose the semantic features of NPO, allowing for the identification of key patterns and trends within the data:

$$\Delta^{-1}(s_k, a_k) = K + a_k = \beta. \tag{5}$$

This transformation enables the model to extract and analyze the keywords and characteristic parameters associated with hotspot information, thereby facilitating the construction of a robust time series model based on big data.

## Semantic feature analysis model

This section presents a explanation of the semantic feature analysis model, which consists of two key components: binary semantic fitting and a semantic similarity model.

### Binary semantic fitting

Binary semantics refers to the representation of semantic features using a pair of elements, such as $(s_k, a_k)$, where $s_k$ denotes a semantic component, and $a_k$ represents its associated attribute. In this study, the semantic feature extraction is based on a word list structured as a binary tree, where each node represents a semantic pair.

To quantify the relationship between two binary semantic pairs, $(s_k, a_k), (s_l, a_l)$, we introduce a distance formula that measures the semantic distance between them:

$$d((s_k, a_k), (s_l, a_l)) = \Delta(|\Delta^{-1}(s_k, a_k) - \Delta^{-1}(s_l, a_l)|). \tag{6}$$

The distance metric is crucial for understanding how closely related different semantic pairs are within the NPO context.

To retrieve relevant semantic features efficiently, the model employs a closed frequent item retrieval method. This method identifies frequent binary semantic pairs from the NPO data, which are essential for capturing the core sentiments within the network:

$$
\begin{aligned}
(\bar{s}, \bar{a}) &= \varphi_1((s_1, a_1), (s_2, a_2), \cdots, (s_n, a_n)) \\
&= \Delta\left(\sum_{j=1}^{n} \frac{1}{n} \Delta^{-1}(s_j, a_j)\right).
\end{aligned}
\tag{7}
$$

Here: $(\bar{s}, \bar{a})$ represents the average semantic feature derived from a set of binary semantic pairs. $\varphi_1$ is the retrieval function that aggregates and transforms these pairs.

This method allows for the identification of frequent semantic elements, which are then organized into a tree-like thesaurus structure, as shown in Fig. 1. The thesaurus helps in understanding the distribution of semantic features across different topics within NPO.

The self-feature sequence and decomposition sequence are as follows:

$$
\begin{aligned}
P &= \{p_1, p_2, \cdots, p_n\} \\
Q &= \{q_1, q_2, \cdots, q_m\}.
\end{aligned}
\tag{8}
$$

These sequences represent the core semantic features and their decomposed elements, respectively. The binary semantic fitting process is thus realized by statistically analyzing the cross-item information chain, which estimates the state of semantic features across different contexts in the NPO data.

### Semantic similarity model

The semantic similarity model utilizes a twin neural network architecture to measure the similarity between sentences within the NPO data. This model is built upon the Bi-LSTM network and an attention mechanism to comprehensively extract semantic features from sentences.

The twin neural network consists of two identical sub-networks, each responsible for processing one of the two sentences in a pair. The sentences are converted into fixed-length feature representation vectors, which are then compared to evaluate their similarity. Importantly, the parameters of the two sub-networks are shared, ensuring consistent feature extraction.

The model includes several key layers:

Input layer: Accepts the sentence pair for analysis.

Word embedding layer: Converts words into dense vectors that represent their meanings.
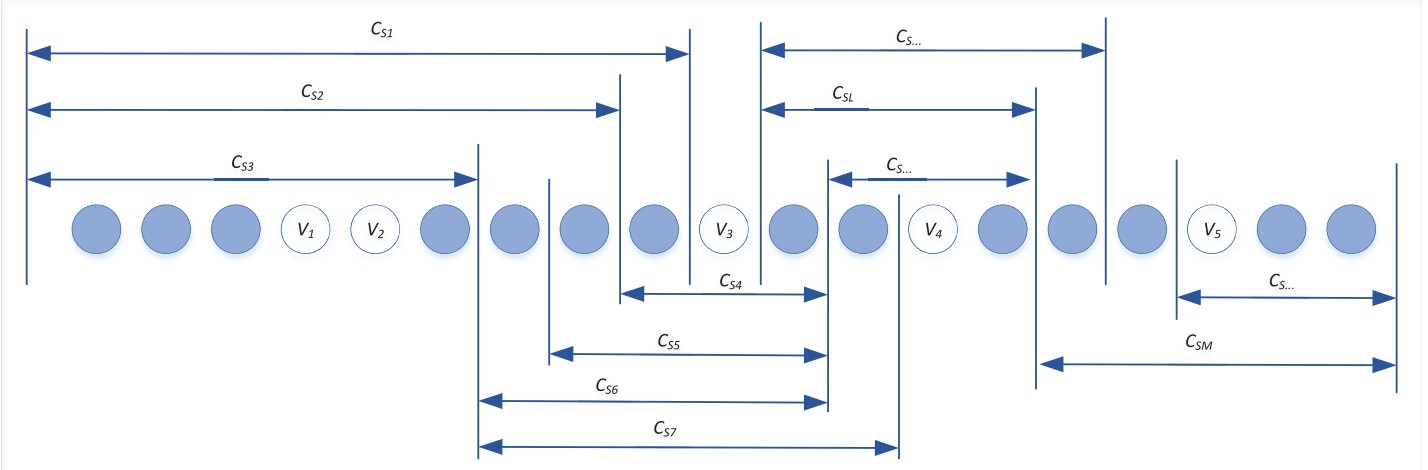

**Figure 1 Structure chart of NPO thesaurus.**

Bi-LSTM layer: Captures the sequential dependencies in the sentence by processing the word vectors in both forward and backward directions.

Attention layer: Highlights the most important words in each sentence, enhancing the model's focus on critical semantic features.

Matching layer: Calculates the similarity between the two sentence vectors using various distance metrics.

Fully Connected layer: Combines the extracted features and refines the similarity score.

Output layer: Produces the final similarity score between the sentences.

The first Bi-LSTM contains 128 hidden layers, and the second Bi-LSTM contains 32 hidden layers. The model uses a reusable Bi-LSTM neural network, that is, for two input sentences, their Bi-LSTM neural network weights are shared. Assuming that the neural network of layer L in the stacked neural network is represented by $H_l$, then at time $t$, the semantic feature values of sentences output by the neural network of this layer are:

$$h_t^l = H_l\left(x_t, h_{t-1}^l\right)$$
$$x_t^l = h_t^{l-1}$$

$$(9)$$

The forward LSTM is in accordance with the word sequence $x = [x_1, x_2 \ldots, x_n]$ The feature extraction of the word vector matrix is carried out in the forward direction; on the contrary, the backward LSTM extracts the word vector matrix corresponding to the sentence sequence from the back to the front.

To measure the similarity between the two sentences, represented by vectors $s = [s_1, s_2, \ldots, s_n]$ and $p = [p_1, p_2, \ldots, p_n]$ of two sentences, the model calculates their Manhattan distance, cosine similarity and Euclidean distance, respectively.

**Manhattan distance** calculates the sum of the absolute differences between corresponding elements of the two sentence vectors. It is particularly effective in cases

where differences in individual dimensions are more important than their overall orientation. Manhattan distance is defined as:

$$\text{dist}_{\text{man}}(s, p) = \sum_{i=1}^{n} |s_i - p_i|. \tag{10}$$

Cosine similarity measures the cosine of the angle between the two sentence vectors, effectively assessing how closely aligned the vectors are in multidimensional space. It is particularly useful when the magnitude of the vectors (*i.e.*, the length of the sentences) is less important than their direction (*i.e.*, their overall semantic orientation). Cosine similarity is defined as:

$$\text{dist}_{\text{cosine}}(s, p) = \frac{\sum_{i=1}^{n}(s_i * p_i)}{\sqrt{\sum_{i=1}^{n}(s_i)^2} * \sqrt{\sum_{i=1}^{n}(p_i)^2}}. \tag{11}$$

**Euclidean distance** calculates the straight-line distance between the two vectors in multidimensional space, reflecting the overall difference between them. Unlike Manhattan distance, which sums the absolute differences, Euclidean distance considers the square root of the sum of squared differences, making it sensitive to larger discrepancies between individual components. Euclidean distance is defined as:

$$\text{dist}_{eu}(s, p) = \sqrt{\sum_{i=1}^{n}(s_i - p_i)^2}. \tag{12}$$

By employing all three metrics, the model gains a robust and multidimensional understanding of sentence similarity. The integration of these metrics in the matching layer allows the model to capture the nuanced ways in which sentences can be similar or different, thereby enhancing the accuracy of semantic similarity predictions.

The fully connected layer then combines these features nonlinearly to refine the final similarity score, resulting in a comprehensive evaluation of how closely related the two sentences are within the context of NPO data.

The function of the whole connective layer is to make nonlinear combination of features and extract the relationship between features, so as to better distinguish the semantic similarity between sentences. First of all, the output vectors of Eqs. (1) and (2) of the model are matched in one-dimensional representation, as shown in Formula (13).

$$r = \text{concatenate}\,(s, p, \mathit{dist}_{man}, \mathit{dist}_{cosine}, \mathit{dist}_{eu}). \tag{13}$$

Then, for the obtained vector R, the model uses three fully connected layers, and selects ReLU function as the activation function to combine R. Its dimension changes are shown in Fig. 2. After the full connection layer, the dimension of feature vector r gradually

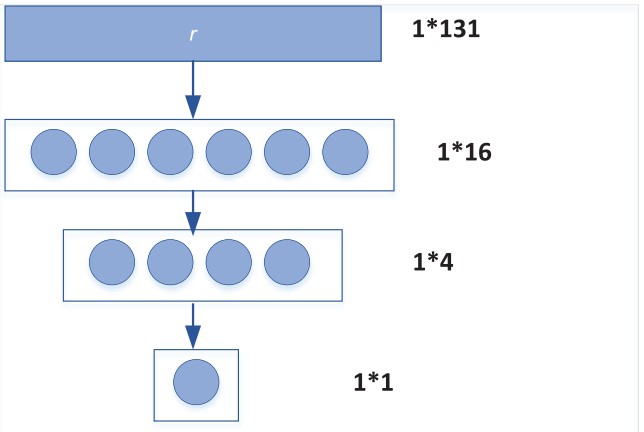

**Figure 2** **Transformation of full connection layer.**

changes from 1 * 131 to 1 * 16, 1 * 4, and finally becomes a 1 * 1 vector, which is the sentence similarity value of the final output of the model.

## Calculation of emotional intensity

In the context of an NPO event, Internet users express their opinions through blog posts on various platforms. These posts often receive comments, likes, and views, reflecting the public's emotional response to the event. Understanding the emotional intensity within these interactions is crucial for gauging public sentiment. This section outlines the method for calculating the emotional intensity associated with such events.

The emotional intensity, denoted by μ, is governed by the first law of emotional intensity, expressed as:

$$\mu = K\log(1 + \Delta p) \tag{14}$$

where $K$ is the strength coefficient, a parameter that scales the emotional response. The value of $K$ depends on the context and the sensitivity of the audience to the event. $\Delta p$ is the difference of value rate, which quantifies the change in public opinion before and after the event. This parameter is crucial as it captures the event's impact on public sentiment. A higher absolute value of $\Delta p$ indicates a more significant shift in opinion, implying a greater emotional impact.

To better understand how these parameters influence emotional intensity, consider a scenario where an NPO event initially receives neutral feedback, but over time, the public sentiment becomes overwhelmingly positive. The value of $\Delta p$ would reflect this shift, and when multiplied by the strength coefficient $K$, the resulting emotional intensity $\mu$ would indicate the degree of public engagement with the event.

The emotional intensity of an online public opinion event $E_i$, is derived from the aggregation of emotional content within blog posts, comments, likes, and views. The process involves the following steps:

Gathering data: Collect all relevant texts, including blog posts and associated comments, from various social platforms.

Quantifying emotion: For each text $s_j$, assign an emotional value based on its content. This value reflects the sentiment expressed, whether positive, negative, or neutral.

Averaging emotional values: Compute the average emotional value across all texts related to the event:

$$S_{\text{event}}(E_i) = \frac{\sum_{j=1}^{N} s_j}{N}, j \in [1, N] \tag{15}$$

where N is the total number of texts and comments involved in public opinion events.

Incorporating Emotional Intensity: The comprehensive emotional index $SE(E_i)$ is then calculated by multiplying the average emotional value by the emotional intensity $\mu_i$:

$$SE(E_i) = S_{\text{event}}(E_i) \cdot \mu_i. \tag{16}$$

The comprehensive emotional index $SE(E_i)$ not only quantifies the emotional response but also contextualizes it within the broader narrative of public opinion.

## Judicial judgment process

The sourcing of events within the judicial judgment system is accomplished through two primary methods: manual addition and automatic detection. These methods work together to ensure that relevant public opinion events are accurately identified and entered into the event database.

Manual addition process: In the manual addition process, users actively participate by identifying events of interest. If a user is interested in a particular public opinion event, they first search for the event in the event database. If the event is not found, the user can manually add it. This method allows for precise control over the events included in the system, ensuring that specific topics of interest, such as those related to public security or legal matters, are tracked and monitored.

Automatic detection and addition process: Complementing the manual method, the system also automatically identifies and adds events by regularly crawling the hot topic lists of major websites such as Weibo, Baidu, Sogou, and Zhihu. The topics gathered through this web scraping process are temporarily stored in a topic database. The system then applies an intelligent algorithm to evaluate these topics against predefined criteria to determine their relevance to the user.

Algorithm and decision-making process: The algorithm is designed to filter topics based on user-defined interests, such as public security or legal issues. It assesses each topic to decide whether it should be added to the event database. If a topic meets the criteria, the system automatically reports it to the event database, creating a new entry with the topic as the event name. Topics deemed irrelevant are marked accordingly, and the system continues to the next topic in the queue. This decision-making process ensures that only pertinent events are tracked, optimizing the database for user needs.

Scheduler process and workflow integration: The entire workflow is orchestrated by a scheduler process that runs continuously in the background. This scheduler ensures that both manual and automatic additions are integrated smoothly into the system,

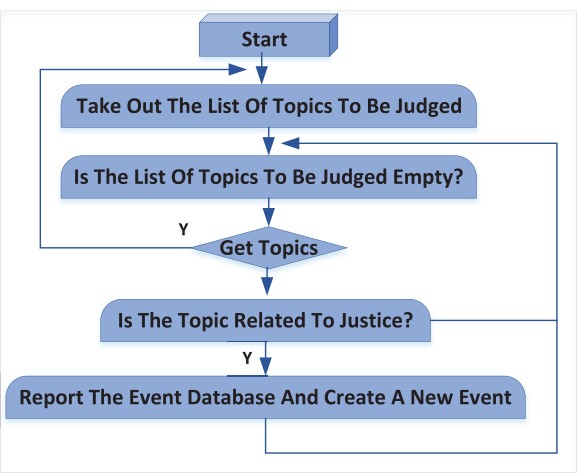

**Figure 3  Judicial judgment process.**       

maintaining an up-to-date and relevant event database. The process is cyclical, allowing for ongoing detection and database updating.

To better illustrate the workflow, Fig. 3 provides a visual representation of the judicial judgment process. This flow chart maps out the steps from initial event detection—whether manual or automatic—through to the final database entry, offering a clear and concise guide to the system's operation.

# EXPERIMENT AND ANALYSIS

## Experimental settings

The proposed algorithm is verified by using the liar data set (https://zenodo.org/records/8214425, doi: 10.5061/dryad.8w9ghx3q2) which is the benchmark data set for forgery news detection. In order to simulate the outbreak of online public opinion in real life, we conducted a five fold cross validation.

## Model training results

The cross entropy is used as the loss function of the model; The Adam optimizer is selected as the optimization algorithm. In the whole training process, the weight of the pre training word vector is constantly updated, and the other parameters are updated by batch processing method. The training results are shown in Fig. 4.

It can be found that the accuracy of the test set is not much different from that of the training set, and the loss value of the test set is also consistent with the loss value of the training set, indicating that the model can be applied in practice.

The training results of different models on the test set are shown in Fig. 5.

Word Mover's Distance (WMD) and Continuous Bag of Words (CBOW) are traditional models for calculating sentence semantic similarity, yet their performance is often inferior to that of more advanced deep learning algorithms. Both the convolutional neural network (CNN) and bidirectional long short-term memory (Bi-LSTM) models employ a twin neural network structure. This structure extracts semantic features from input sentences using either CNN or Bi-LSTM architectures, followed by matching

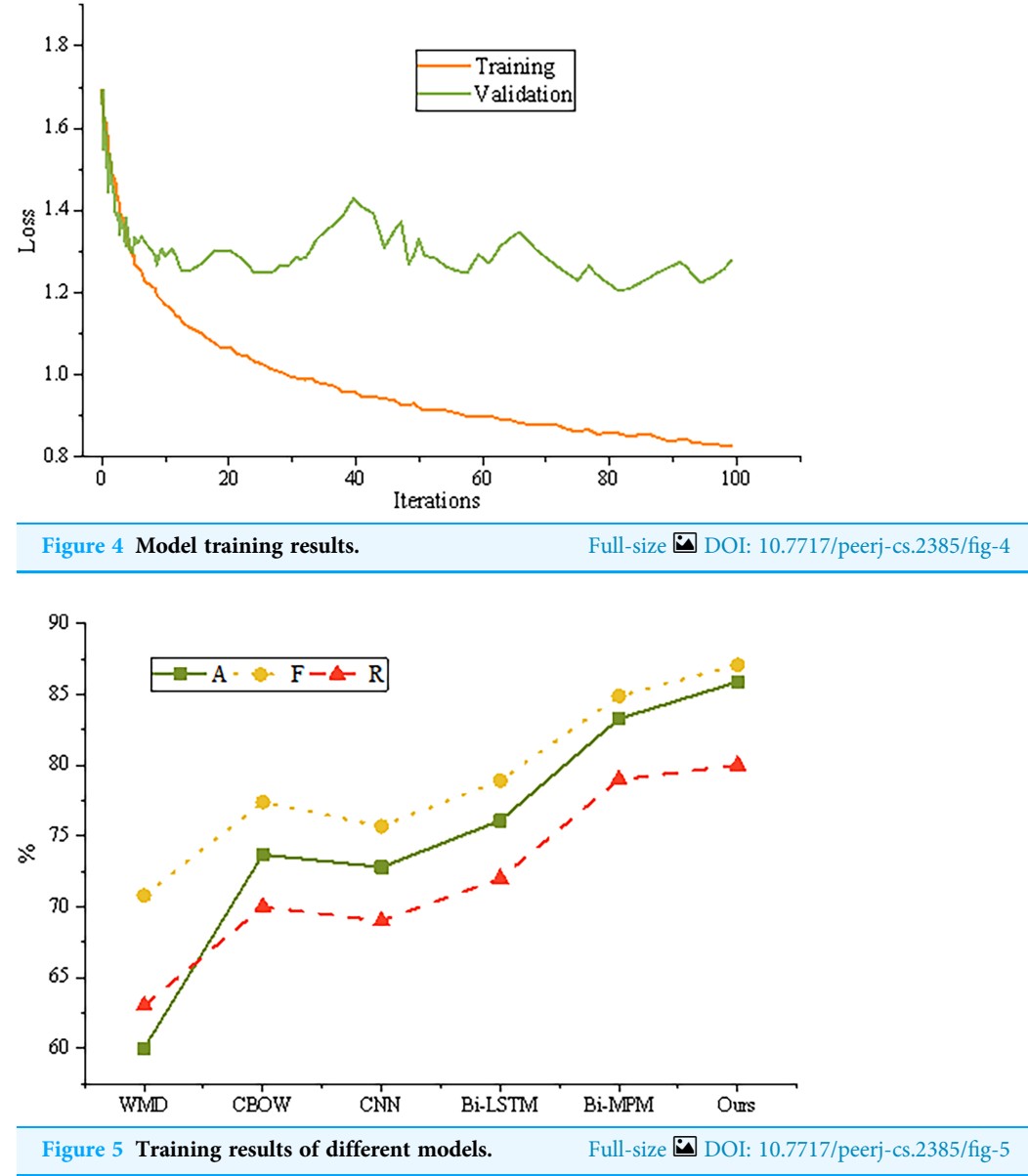

**Figure 4  Model training results.**

**Figure 5  Training results of different models.**

decisions based on the sentence vectors. Among these, the Bi-LSTM model demonstrates superior performance compared to CNN, largely due to its recurrent neural network design, which is particularly adept at handling the sequential nature of sentence structures. The Bi-LSTM model facilitates multi-angle sentence similarity matching, wherein each step of a sentence is aligned with all time steps of another sentence. The results of these alignments are transformed into multiple vectors *via* a CNN or LSTM network, and are subsequently combined to yield the final similarity score.

However, the validity and reliability of the proposed emotional intensity measures, which build upon these models, require further discussion. While WMD and CBOW provide a foundational understanding of semantic similarity, their limitations highlight the need for more robust methods like Bi-LSTM in capturing the nuances of emotional

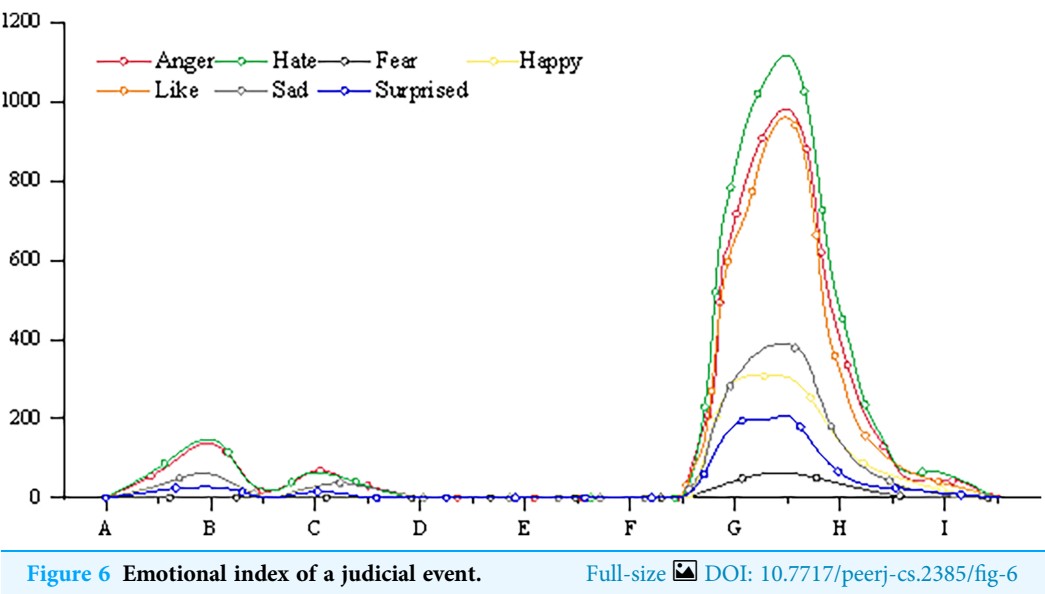

**Figure 6 Emotional index of a judicial event.**

responses. Existing methods for measuring emotional responses in online content typically rely on sentiment analysis tools, which often struggle with contextual understanding and intensity quantification. The choice of Bi-LSTM, in this context, is justified due to its ability to consider the sequential dependencies within sentences, making it more capable of capturing the subtleties of emotional expression. This approach not only aligns with current advancements in natural language processing but also ensures a higher degree of validity and reliability in the measurement of emotional intensity in online content.

### Emotional analysis of NPO events in the judicial domain

The emotion of individual or group affects the guidance of public opinion. Import a specific criminal case from the database for emotional analysis, and the results are shown in Fig. 6.

It can be seen that the event has experienced three ups and downs of public opinion before and after the event, and emotional analysis can further show the netizens' emotions about the event. The Public Security Bureau handed over the suspect to the people's Procuratorate. The public opinion peaks caused by these two behaviors are mainly anger and disgust, which expresses the dissatisfaction of netizens to the public inspection department in the region. When the Supreme People's Procuratorate ordered to cancel the decision of non-prosecution, public opinion once again set off a climax. However, different from the previous two high tides, the proportion of "like" emotion and "anger" and "disgust" in public opinion this time is similar. Obviously, netizens "angry" is the behavior of the public inspection department, but "like" is the decision of the Supreme People's Procuratorate.

## DISCUSSION

The temporal dependence of the temporal information of another mode is not taken into account. The proposed semantic similarity model is based on twin neural network. It

extracts sentence features comprehensively and effectively by using double stack Bi LSTM and integrating attention mechanism. Finally, the full connection layer is used to output the semantic similarity value between sentences. The accuracy rate of the proposed model is 85.9% and the F1 value is 87.1 in the test set, which is better than the existing sentence semantic similarity models. It shows that the model is effective and feasible in the calculation of sentence semantic similarity.

Emotional analysis can effectively reflect the different positions of public opinion. Because the vast majority of public opinion events related to topics such as the public security organs, procuratorial organs and law enforcement organs can grasp the position of public opinion in public opinion through emotional analysis, so as to better interpret, popularize and guide public opinion. In addition, training and testing the model on the existing public annotation data set, predicting the emotion of the massive microblog comments in the system, helps the judicial department to depict the development trend of public opinion, and can also reveal the views and attitudes of public opinion participants more deeply.

# IMPLICATIONS

## Theoretical implications

Although the traditional method of sentence semantic similarity calculation is effective, it needs to construct feature engineering manually. Ignored at the same time, the traditional methods for the use of domain knowledge, law, there is a special public opinion transmission based on social media, so this article from the perspective of legal elements to combine domain knowledge into text vector said, built by the judicial field of public opinion monitoring model based on semantic feature analysis, by using S-Bi-LSTM network in the judicial field of semantic similarity computation, Binary semantic fitting and calculation of public emotion intensity, sentence semantic similarity model is put forward in this article based on the twin neural network structure, by using double stacked Bi-LSTM and fusion mechanism of attention to comprehensive and effective sentence features are extracted, and then the matching layer for the interaction between the two sentence information, finally use full connection layer to the various characteristics of combination operation, and output the semantic similarity value between sentences.

The results show that the prediction accuracy of the algorithm is high, which is helpful to provide a theoretical basis for intelligent supervision in the judicial field.

## Practical implications

Due to the lack of unified method for handling public opinion. The experimental results show that the proposed monitoring model in the judicial field based on semantic feature analysis has good experimental effect. Therefore, we can specifically design a public opinion monitoring system for the judicial field.

The public opinion risk assessment system of judicial cases designed in this article adopts B/S mode structure, users only need to have a browser to complete the use, no need to configure the client, and the hardware requirements of users are low. The system uses a

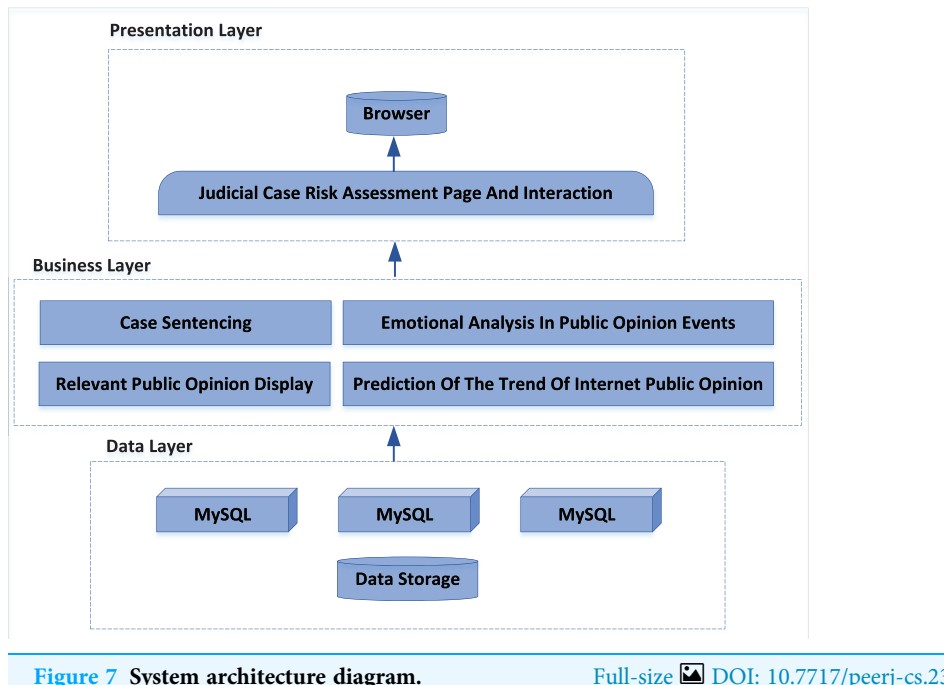

**Figure 7  System architecture diagram.**     

three-tier architecture of presentation layer, business layer and data layer, and each layer is connected through interfaces. The system architecture is shown in Fig. 7.

Presentation layer: the presentation layer can provide a convenient and comprehensive user interface for browsers, which is used to display information and provide an interface for users to input data. It can also execute some simple business logic through some script languages. Through the use of visual form, the presentation layer realizes the interaction between the user and the system, and the user communicates with the server by submitting the request, so as to obtain the processing results. Bootstrap is an opensource web front-end framework, which is developed on the basis of HTML5 and CSS3 technology, has a powerful component library and supports a variety of browsers. Bootstrap is simple, easy to use and fast to develop. This article uses the bootstrap framework to write the presentation layer.

Business layer: all requests from the presentation layer will be delivered to the business layer for processing. The latter will send the processing results to the designated place according to the programming function, which can be returned to the presentation layer and may trigger database operations. As the core of the system, the business layer includes four functional modules: "display of relevant public opinion", "case punishment", "emotional analysis in public opinion events", "NPO trend prediction". Flask is a web framework written in Python language, which has strong scalability and compatibility. It can quickly implement a website or web service using Python language. This article uses the flash framework to write the business layer, which can meet the calculation and logic requirements of all functional modules of the system, and feedback the processing results by using the interface.

Data layer: the public opinion monitoring system takes "event" as the basic unit, and the event is defined as the topic to be monitored. Each entry in the event database corresponds to an event, and stores other information of the event, including some foreign keys (related event list, news list, microblog list), basic information (time, place, person, organization, *etc.*), front-end mapping information (forwarding network relationship diagram data, sentiment analysis curve graph data, user type pie chart data, *etc.*).

## CONCLUSION

In this research, a semantic similarity calculation model based on S-Bi-LSTM is constructed to realize supervision and prediction of NPO. The training results on the liar data set show that the model performs well in A, R and F1, with high prediction accuracy and strong real-time performance. It is of extraordinary importance to improve the judicial department's NPO supervision ability, and can more deeply reveal the views and attitudes of public opinion participants. Based on the verification results of the model, this article also puts forward the application scenario of the judicial case public opinion supervision system. The model could find similar cases from the legal elements of the level, effectively improve the judicial regulation ability of network public opinion and greatly alleviate the public relations pressure of judicial organs, so that judicial organs more fair and equitable exercise of power.

## ACKNOWLEDGEMENTS

The author would like to thank the anonymous reviewers for their valuable comments on this article.

### Funding

This work is funded by "Legal research project of Shanxi Law Society", the project number is SXLS2023B11. The funders had no role in study design, data collection and analysis, decision to publish, or preparation of the manuscript.

### Grant Disclosures

The following grant information was disclosed by the authors:
"Legal Research Project of Shanxi Law Society": SXLS2023B11.

### Competing Interests

The author declares that they have no competing interests.

### Author Contributions

- Heng Guo conceived and designed the experiments, performed the experiments, analyzed the data, performed the computation work, prepared figures and/or tables, authored or reviewed drafts of the article, and approved the final draft.

## Data Availability

The data is available at Zenodo and Dryad:

-Kasuga, H., Endo, S., Masuishi, Y., Hidaka, T., Kakamu, T., & Fukushima, T. (2023). Data from: Public opinion in Japanese newspaper readers' posts under the prolonged COVID-19 infection spread 2019-2021: Contents analysis using Latent Dirichlet Allocation [Data set]. Zenodo. https://doi.org/10.5061/dryad.8w9ghx3q2.

-Kasuga, Hideaki; Endo, Shota; Masuishi, Yusuke et al. (2023). Data from: Public opinion in Japanese newspaper readers' posts under the prolonged COVID-19 infection spread 2019-2021: Contents analysis using Latent Dirichlet Allocation [Dataset]. Dryad. https://doi.org/10.5061/dryad.8w9ghx3q2.

## Supplemental Information

Supplemental information for this article can be found online at http://dx.doi.org/10.7717/peerj-cs.2385#supplemental-information.

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
