# Peer review of "Design of judicial public opinion supervision and intelligent decision-making model based on Bi-LSTM"

_PeerJ Computer Science, doi:10.7717/peerj-cs.2385_

## Round 0.1 · original submission · Major Revisions

Dear authors

Based on the experts opinion on your manuscript, I'm happy to inform you that your manuscript is above the border line but not acceptable in it's current form. Therefore, we advise you to revise the manuscript in light of the comments of the experts and mine below.

Please submit a detailed rebuttal along with the updated manuscript.

1.why the study is important and who are potential beneficiaries of the study.
2. Clearly state the problem statement of the proposed study.
3. Justify how your model is outperforming that state of the art.
4. Please improve The language of the paper

Reviewer 1 ·

Basic reporting

There are several points that could be improved for clarity, conciseness, and academic rigor. Below are some academic paragraphs highlighting the main issues and suggesting improvements:

(1) In Section 3.1, "Time Series Model," the explanation of the nonlinear feature sequence is vague and lacks a clear introduction to the significance of the variables used. The random variable UiU_iUi and the monitored network public opinion UUU are introduced without sufficient context or explanation of their relevance. The feature extraction and cluster analysis process needs more detail to provide a comprehensive understanding of the methods employed. Additionally, the notation α\alphaα and uuu for baud rate and monitoring frequency response should be better contextualized within the framework of network public opinion (NPO). It would be beneficial to clarify the role of the inter-domain association characteristics and how they contribute to the overall model. The mathematical expressions, while precise, would benefit from a more intuitive explanation to guide the reader through the logic and purpose of each step.

Experimental design

(2) The semantic feature analysis model in Section 3.2 lacks coherence in presenting the binary semantic fitting and similarity model. The binary semantic fitting section should start with a clearer introduction of the binary semantics concept and its relevance to the study. The distance formula provided needs an accompanying explanation of the variables and their significance in the context of semantic feature extraction. The frequent item retrieval method is introduced abruptly; a smoother transition explaining its necessity and application would enhance comprehension. The semantic similarity model, which employs a twin neural network, requires a more detailed explanation of the Bi-LSTM network and attention mechanism. The description of the network structure and layers should be more explicit, detailing how each layer contributes to the overall model performance. Additionally, the model's calculation methods for Manhattan distance, cosine similarity, and Euclidean distance need further clarification to ensure the reader understands their importance in measuring sentence similarity.

Validity of the findings

(3) The section on "Calculation of Emotional Intensity" in 3.3 presents an interesting concept but lacks depth in explaining the underlying principles. The introduction of the first law of emotional intensity and the formula μ=Klog⁡(1+Δp)\mu = K \log(1 + \Delta p)μ=Klog(1+Δp) should be preceded by a more thorough explanation of the parameters involved, particularly the strength coefficient KKK and the difference of value rate Δp\Delta pΔp. The description of how emotional intensity relates to the impact of NPO events is somewhat abstract and could benefit from concrete examples. The process of calculating the text emotion intensity for online public opinion events needs to be more detailed, with a clear step-by-step explanation of how the values are derived and aggregated. Furthermore, the comprehensive emotion index formula SE(Ei)=Sevent(Ei)⋅μiSE(E_i) = S_{event}(E_i) \cdot \mu_iSE(Ei)=Sevent(Ei)⋅μi should be contextualized within a broader discussion of its implications for understanding public sentiment.

Additional comments

(4) In the final section, "Judicial Judgment Process" (3.4), the two methods for sourcing events are presented in a disjointed manner. The description of manual and automatic addition processes should be integrated into a cohesive narrative that explains the workflow from event detection to database entry. The explanation of the algorithm used to identify topics of interest is too brief and lacks specifics about the criteria and decision-making process. Additionally, the flow chart mentioned (Figure 3) should be referenced more explicitly in the text, providing a visual guide to support the procedural description. Enhancing this section with more detailed explanations of the scheduler process and its role in maintaining the event database would provide a clearer understanding of the system's functionality.

·

Basic reporting

The manuscript presented has several shortcomings that could be improved for clarity, coherence, and academic rigor.
Te introduction lacks a clear and concise definition of key terms such as "emotional intensity" and "NPO event." Defining these terms at the beginning would provide a clearer understanding for readers. Additionally, the description of how Internet users express their opinions through blog posts is vague. It would be beneficial to specify the platforms and types of interactions (e.g., comments, likes, shares) that are considered in the analysis

Experimental design

The formula for the first law of emotional intensity, μ=Klog(1+Δp), is introduced abruptly without sufficient context or explanation. The parameters K and Δp should be clearly defined, and the rationale for using this formula should be discussed. Furthermore, it is important to explain the theoretical basis for the logarithmic function and how it captures the emotional intensity

Validity of the findings

The explanation of how the emotional intensity of an entire NPO event is calculated is somewhat disjointed. The text should clearly outline the steps involved in this process, including how individual emotional intensities (s_j) are aggregated. The notation used in the formula S_event (E_i) is inconsistent and could be clarified. Additionally, the significance of the comprehensive emotion index (SE(E_i)) should be elaborated upon, explaining its role in assessing the overall emotional response to an event

Additional comments

The text lacks a discussion on the validity and reliability of the proposed emotional intensity measures. It would be useful to include a brief review of existing methods for measuring emotional responses in online content and justify the choice of the current approach.

The introduction could benefit from a more formal academic tone. Phrases such as "netizens' satisfaction with the social governance" should be rephrased to maintain a professional tone. Additionally, the text should be structured to flow logically, starting with the broader context of studying emotional intensity in NPO events, followed by the detailed methodology, and concluding with the significance and potential applications of the research findings

---

## Round 0.2 · accepted · Accept

Based on the input from experts, I'm pleased to inform you that your manuscript is being recommended for publication.

Thank you for your contribution

Reviewer 1 ·

Basic reporting

No comment

Experimental design

No comment

Validity of the findings

no comment

·

Basic reporting

I have reviewed the entire paper and found that no further updates are needed. One reviewer’s comments have been adequately addressed, so there are no conflicts. The paper is acceptable from my side.

Experimental design

No comments

Validity of the findings

No Comments

Additional comments

I have reviewed the entire paper and found that no further updates are needed. One reviewer’s comments have been adequately addressed, so there are no conflicts. The paper is acceptable from my side.